# Hematopoietic Cell Transplantation Trends and Outcomes in Canada: A Registry-Based Cohort Study

Matthew D. Seftel [1,2,*], Ivan Pasic [3,4], Gaganvir Parmar [1,4], Oliver Bucher [5], David S. Allan [1,6], Sita Bhella [3,4], Kevin Anthony Hay [2,7,8], Oluwaseun Ikuomola [5], Grace Musto [5], Anca Prica [3,4], Erin Richardson [5], Tony H. Truong [9] and Kristjan Paulson [10,11]

1  Canadian Blood Services, Vancouver, BC V6H 2N9, Canada; gaganvir.parmar@mail.utoronto.ca (G.P.); daallan@toh.ca (D.S.A.)
2  Division of Hematology, Department of Medicine, Faculty of Medicine, University of British Columbia, Vancouver, BC V5Z 1M9, Canada; kevin.hay@ucalgary.ca
3  Hans Messner Allogeneic Blood and Marrow Transplantation Program, Division of Medical Oncology and Hematology, Princess Margaret Cancer Centre, University Health Network, Toronto, ON M5G 2M9, Canada; ivan.pasic@uhn.ca (I.P.); sita.bhella@uhn.ca (S.B.); anca.prica@uhn.ca (A.P.)
4  Faculty of Medicine, University of Toronto, Toronto, ON M5G 2M9, Canada
5  Department of Epidemiology, CancerCare Manitoba, Winnipeg, MB R3A 1M5, Canada; obucher@cancercare.mb.ca (O.B.); oikuomola@cancercare.mb.ca (O.I.); gmusto@cancercare.mb.ca (G.M.); erichardson@cancercare.mb.ca (E.R.)
6  Department of Medicine and Biochemistry, Microbiology & Immunology, Faculty of Medicine, University of Ottawa, Ottawa, ON K1H 8L6, Canada
7  Division of Hematology, Department of Medicine, University of Calgary, Calgary, AB T2N 1N4, Canada
8  Terry Fox Laboratory, British Columbia Cancer Research Institute, Vancouver, BC V5Z 1L3, Canada
9  Division of Pediatric Hematology/Oncology, Department of Pediatrics, University of Calgary, Calgary, AB T3B 6A8, Canada; tony.truong@ahs.ca
10 Cell Therapy and Transplant Canada, Winnipeg, MB R3E 0V9, Canada; kpaulson@cancercare.mb.ca
11 Department of Medical Oncology and Haematology, CancerCare Manitoba, Winnipeg, MB R3E 0V9, Canada
*  Correspondence: matthew.seftel@blood.ca; Tel.: +1-604-707-3414

**Abstract: Background:** Hematopoietic cell transplantation (HCT) is an established therapy for hematologic malignancies and serious non-malignant blood disorders. Despite its curative potential, HCT is associated with substantial toxicity and health resource utilization. Effective delivery of HCT requires complex hospital-based care, which limits the number of HCT centres in Canada. In Canada, the quantity, indications, temporal trends, and outcomes of patients receiving HCT are not known. **Methods:** A retrospective cohort study of first transplants reported to the Cell Therapy Transplant Canada (CTTC) registry between 2000 and 2019. We determined overall survival (OS) and non-relapse mortality (NRM), categorizing the cohort into early (2000–2009) and later (2010–2019) eras to investigate temporal changes. **Results:** Of 18,046 transplants, 7571 were allogeneic and 10,475 were autologous. Comparing the two eras, allogeneic transplants increased in number by 22.3%, with greater use of matched unrelated donors in the later era. Autologous transplants increased by 10.9%. Temporal improvements in NRM were observed in children and adults. OS improved in pediatric patients and in adults receiving autologous HCT. In adults receiving allogeneic HCT, OS was stable despite the substantially older age of patients in the later era. **Interpretation:** HCT is an increasingly frequent procedure in Canada which has expanded to serve older adults. Noted improvements in NRM and OS reflect progress in patient and donor selection, preparation for transplant, and post-transplant supportive care. In allogeneic HCT, unrelated donors have become the most frequent donor source, highlighting the importance of the continued growth of volunteer donor registries. These results serve as a baseline measure for quality improvement and health services planning in Canada.

**Keywords:** hematopoietic cell transplant; cellular therapy; health outcomes

## 1. Introduction

Hematopoietic cell transplantation (HCT) is an established therapeutic procedure for hematological malignancies and serious non-malignant hematological disorders [1,2]. It encompasses two major modalities: autologous HCT, which involves administering high doses of myelotoxic chemo- or radiotherapy followed by the reinfusion of autologous hematopoietic progenitor cells to recover the patient's hematopoietic system, and allogeneic HCT, which includes myelotoxic or immunosuppressive therapy followed by infusion of hematopoietic progenitor cells derived from a related donor, volunteer unrelated donor, or placental cord blood donor [3].

Despite its curative potential, HCT is a technically complex and resource-intensive procedure, often associated with considerable toxicity. In Canada, HCT is offered by a limited number of specialized transplant centres, all based at major hospitals. Over time, advances in donor selection [4], pretransplant therapy, post-transplant supportive care [5–7], and graft-versus-host disease (GVHD) management [8,9] have influenced HCT trends and outcomes. Several national [10] and international [11] registries have described HCT trends and outcomes and demonstrated improvements in clinical outcomes over time [12,13]. However, the specific quantity, trends, and clinical outcomes of HCT over time in Canada are not known.

To address this knowledge gap, we examined the characteristics of HCT in Canada using registry data. Our primary aim was to analyze the trends in HCT procedures and characteristics of recipients, as we hypothesized that the HCT landscape has evolved over time in line with other developed countries. Additionally, we examined the hypothesis that survival among transplant recipients have improved over time, potentially attributable to advances in supportive care leading to a reduction in HCT-related toxicities and associated non-relapse mortality.

## 2. Methods

### 2.1. Study Design, Participants and Setting

We conducted a retrospective cohort analysis of autologous and allogeneic HCTs at Canadian transplant centres performed between 1 January 2000, and 31 December 2019. To determine whether patient characteristics have changed and whether outcomes have improved over time, patients were grouped into two periods: 2000 to 2009 (early era) and 2010 to 2019 (later era). We included children and adults who received a first HCT at any Canadian transplant centre reporting to the Cell Therapy Transplant Canada (CTTC) registry over this period.

Separate analyses of clinical outcomes were performed on patients with non-malignant disease indications. Additionally, a subgroup analysis of allogeneic HCTs was performed on adults with indication of acute leukemia: acute myeloid leukemia (AML), acute lymphoblastic leukemia (ALL), acute leukemia of ambiguous lineage, myelodysplastic syndrome (MDS) and myeloproliferative neoplasia (MPN). This subgroup analysis was intended to remove the potential confounding effect of HCT indications which were more frequently performed in the earlier HCT era and whose clinical outcomes may differ substantially from the other allogeneic HCT indications.

### 2.2. Data Sources

Through a collaboration between CancerCare Manitoba and CTTC, patient, donor, and outcome data from these centres were voluntarily collected and analyzed. The CTTC registry is a research consortium consisting of 15 adult and pediatric HCT centres across Canada who report their transplant data to the Centre for International Blood and Marrow Transplant Research (CIBMTR), Milwaukee, WI. These Canadian-specific data are then transferred by CIBMTR to the CTTC registry and housed by the CTTC registry at Cancer-Care Manitoba. Regular central auditing of the data is performed to ensure consistency and quality. Transplant essential data (TED) forms prior to transplantation; at 100 days, 6 months, and 1-year post-transplant; and annually thereafter are available for analysis.

Socio-demographic data and medical information regarding diagnosis, treatment, transplant, and follow-up were collected. Data on donor types, cell sources, disease progression or relapse, and mortality were also recorded.

### 2.3. Outcomes

Trends in the patient characteristics, transplant type, and indication of HCTs in Canada were assessed as primary outcomes. Clinically relevant outcomes including overall survival (OS) and non-relapse mortality (NRM) were also examined as primary outcomes for HCT recipients over this 20-year period. OS for all patients was defined as the time from HCT until death. NRM was defined as death from any cause other than underlying disease relapse or progression after HCT. All data were censored at the date of last follow-up, date of last contact, date of death or date of second transplant.

### 2.4. Statistical Analysis

Baseline patient, disease indication, and HCT characteristics for first transplants were summarized using descriptive statistics and reported as absolute numbers and proportions for each of the two eras. Statistical significance for comparisons was assessed using Fisher's exact test for categorical data. OS and TRM were analyzed by HCT type (allogeneic or autologous), disease type (malignant or non-malignant), age groups (adult, age $\geq$ 18 years or pediatric, age < 18 years) and era (early or later) for first transplants. Survival probabilities were estimated using the Kaplan–Meier method and survival curves were compared using the log-rank test. NRM was analyzed using cumulative incidences and Fine and Gray's tests to accommodate competing risks of death from relapse/progression. Survival probabilities and cumulative incidences were reported with 95% confidence intervals (CI) and $p$-values < 0.05 were considered statistically significant.

### 2.5. Ethics Approval

All patients included in this study gave written consent to participate in the research database and to have their data included in observational research. This study was approved by the Institutional Review Board of the University of Manitoba.

### 3. Results

A total of 18,046 first HCTs were reported from 1 January 2000 to 31 December 2019 from 15 transplant centres in Canada. Autologous transplants accounted for 10,475 (58%) of first HCTs performed. Patient characteristics for first allogeneic and autologous transplants are presented in Tables 1 and 2, respectively. Median age at HCT increased between the early (2000–2009) and later (2010–2019) eras: for allogeneic HCT, the median age increased from 42 years to 50 years ($p$ < 0.0001), while for autologous transplants, the median age increased from 53 years to 58 years ($p$ < 0.0001). The proportion of older adults (>64 years old) receiving allogeneic HCT increased 5-fold between the early and later eras (1.7% vs. 9.4%, $p$ < 0.0001). The proportion of older adults receiving autologous transplants doubled (10.4% vs. 21.5%, $p$ < 0.0001).

**Table 1.** Baseline population characteristics for first allogeneic transplants. Analysis of patients receiving a first allogeneic HCT between 1 January 2000 and 31 December 2019.

| | Total N (%) | 2000–2009 N (%) | 2010–2019 N (%) | $p$-Value |
|---|---|---|---|---|
| **Total** | 7571 (100) | 3407 (45) | 4164 (55) | <0.0001 |
| **Age at HCT in years (median, range)** | 45 (0–75) | 42 (0–75) | 50 (0 -74) | <0.0001 |
| **Age groups (N, %)** | | | | |
| Pediatric [<18 years] | 1479 (19.5) | 765 (22.5) | 714 (17.2) | <0.0001 |
| Young adult [18–39 years] | 1565 (20.7) | 823 (24.12) | 742 (17.8) | <0.0001 |

| | Total N (%) | 2000–2009 N (%) | 2010–2019 N (%) | *p*-Value |
|---|---|---|---|---|
| Middle-aged adult [40–64 years] | 4078 (53.9) | 1760 (51.7) | 2318 (55.7) | 0.0005 |
| Older adult [65+ years] | 448 (5.9) | 58 (1.7) | 390 (9.4) | <0.0001 |
| *Unknown* | 1 (0.01) | 1 (0.01) | 0 (0) | 0.45 |
| **Sex (N, %)** | | | | |
| Male | 4397 (58.3) | 1987 (58.9) | 2410 (57.9) | 0.3979 |
| Female | 3143 (41.7) | 1389 (41.1) | 1754 (42.1) | 0.3979 |
| *Unknown* | 31 (0.4) | 31 (0.4) | 0 (0) | <0.0001 |
| **Donor type (N, %)** | | | | |
| Matched related | 3386 (44.7) | 1861 (54.6) | 1525 (36.6) | <0.0001 |
| Syngeneic (monozygotic twin) | 50 (0.7) | 34 (1.0) | 16 (0.4) | 0.0014 |
| Haplo-identical | 369 (4.9) | 115 (3.4) | 254 (6.1) | <0.0001 |
| Other relative | 14 (0.2) | 10 (0.3) | 4 (0.1) | 0.0594 |
| Matched unrelated | 3542 (46.8) | 1197 (35.1) | 2345 (56.3) | <0.0001 |
| Mismatched unrelated | 209 (2.8) | 189 (5.6) | 20 (0.5) | <0.0001 |
| Multiple donor | 1 (0.01) | 1 (0.03) | 0 (0) | 0.4426 |
| **Cell source (N, %)** | | | | |
| Bone marrow | 1734 (22.2) | 1043 (30.6) | 691 (16.6) | <0.0001 |
| Peripheral blood | 5304 (68.1) | 2093 (61.4) | 3211 (76.9) | <0.0001 |
| Cord blood | 511 (6.6) | 240 (7.0) | 271 (6.5) | 0.2820 |
| *Missing data* | 42 (0.5) | 42 (1.0) | 0 (0.0) | <0.0001 |
| **Indication for HCT (N, %)** | | | | |
| Acute myelogenous leukemia | 2781 (36.7) | 1093 (32.1) | 1688 (40.5) | <0.0001 |
| Acute lymphoblastic leukemia | 1074 (14.2) | 530 (15.6) | 544 (13.1) | 0.0021 |
| Myelodysplastic/myeloproliferative disorders (+preleukemia) | 912 (12.0) | 376 (11.0) | 536 (12.9) | 0.0158 |
| Non-Hodgkin lymphoma | 720 (9.5) | 428 (12.6) | 292 (7.0) | <0.0001 |
| Chronic myelogenous leukemia | 389 (5.1) | 247 (7.3) | 142 (3.4) | <0.0001 |
| Chronic lymphocytic leukemia | 344 (4.5) | 186 (5.5) | 158 (3.8) | 0.0006 |
| Plasma cell disorder (+multiple myeloma) | 37 (0.5) | 31 (0.9) | 6 (0.1) | <0.0001 |
| Hodgkin lymphoma | 20 (0.3) | 17 (0.5) | 3 (0.1) | 0.0004 |
| Other malignancies | 417 (5.5) | 120 (3.5) | 297 (7.1) | <0.0001 |
| Severe aplastic anemia | 341 (4.5) | 183 (5.4) | 158 (3.8) | 0.0012 |
| Other non-malignant disease | 520 (6.9) | 187 (5.5) | 333 (8.0) | <0.0001 |
| Other | 16 (0.2) | 9 (0.3) | 7 (0.2) | 0.453 |

**Table 2.** Baseline population characteristics for first autologous transplants. Analysis of patients receiving a first autologous HCT between 1 January 2000 and 31 December 2019.

| | Total N (%) | 2000–2009 N (%) | 2010–2019 N (%) | *p*-Value |
|---|---|---|---|---|
| **Total** | 10,475 (100) | 4966 (47.4) | 5509 (52.6) | <0.0001 |
| **Age at HCT in years (median, range)** | 55 (0–81) | 53 (0–81) | 58 (0–79) | <0.0001 |
| **Age groups (N, %)** | | | | |
| Pediatric [<18 years] | 785 (7.5) | 394 (7.9) | 391 (7.1) | 0.11 |
| Young adult [18–39 years] | 1235 (11.8) | 720 (14.5) | 515 (9.4) | <0.0001 |
| Middle-aged adult [40–64 years] | 6755 (64.5) | 3336 (67.2) | 3419 (62.1) | <0.0001 |
| Older adult [65+ years] | 1699 (16.2) | 515 (10.4) | 1184 (21.5) | <0.0001 |
| *Unknown* | 1 (0.01) | 1 (0.02) | 0 (0) | 0.4741 |
| **Sex (N, %)** | | | | |
| Male | 6321 (60.4) | 2956 (59.6) | 3365 (61.1) | 0.1187 |
| Female | 4149 (39.6) | 2005 (40.4) | 2144 (38.9) | 0.1187 |
| *Unknown* | 5 (0.05) | 5 (0.1) | 0 (0) | 0.0239 |

**Table 2.** *Cont.*

|  | Total<br>N (%) | 2000–2009<br>N (%) | 2010–2019<br>N (%) | *p*-Value |
|---|---|---|---|---|
| **Cell source (N, %)** |  |  |  |  |
| Bone marrow | 95 (0.9) | 83 (1.7) | 12 (0.2) | <0.0001 |
| Peripheral blood | 10,349 (98.4) | 4848 (96.9) | 5501 (99.8) | <0.0001 |
| Cord blood | 0 (0) | 0 (0) | 0 (0) | N/A |
| *Missing data* | 70 (0.7) | 70 (1.4) | 0 (0) | <0.0001 |
| **Indication for HCT (N, %)** |  |  |  |  |
| Plasma cell disorder (+multiple myeloma) | 5176 (49.4) | 2303 (46.4) | 2873 (52.2) | <0.0001 |
| Non-Hodgkin lymphoma | 3224 (30.8) | 1515 (30.5) | 1709 (31) | 0.5816 |
| Hodgkin lymphoma | 1016 (9.7) | 575 (11.6) | 441 (8.0) | <0.0001 |
| Acute myelogenous leukemia | 77 (0.7) | 72 (1.4) | 5 (0.1) | <0.0001 |
| Chronic lymphocytic leukemia | 7 (0.1) | 4 (0.1) | 3 (0.1) | 0.7146 |
| Acute lymphoblastic leukemia | 6 (0.1) | 4 (0.1) | 2 (0.04) | 0.432 |
| Myelodysplastic/myeloproliferative disorders (+preleukemia) | 1 (0.01) | 1 (0.02) | 0 (0) | 0.4741 |
| Chronic myelogenous leukemia | 1 (0.01) | 1 (0.02) | 0 (0) | 0.4741 |
| Other malignancies | 868 (8.3) | 454 (0.1) | 414 (7.5) | 0.0606 |
| Other non-malignant disease | 82 (0.8) | 26 (0.5) | 56 (1.0) | 0.0052 |
| Other | 17 (0.2) | 11 (0.2) | 6 (0.1) | 0.2234 |

The donor types for allogeneic HCT are summarized in Table 1. Matched related donors (MRDs) accounted for the largest proportion of donors in the early era (54.6%), while matched unrelated donors (MUDs) accounted for the largest proportion in the later era (56.3%). The decrease in MRDs between the two eras (54.6% early vs. 36.6% later, $p < 0.0001$) was proportionate to the increase in MUDs (35.1% early vs. 56.3% later, $p < 0.0001$). Mismatched unrelated donors (MMUDs) consisted of single-antigen human leukocyte antigen (HLA) mismatches (7/8 or 9/10, depending on the HLA testing methodology used at each centre). The use of MMUDs dropped in the more recent era (5.6% vs. 0.5%, $p < 0.0001$). The use of HLA mismatched related (haploidentical) donors doubled (3.4% in the early era vs. 6.1% in the later era, $p < 0.0001$).

Regarding cell source, peripheral blood stem cells (PBSCs) were the most common source in both allogeneic and autologous transplants (Tables 1 and 2). PBSC use in allogeneic transplants increased from 61.4% to 76.9% ($p < 0.0001$) over the two eras. In contrast, bone marrow for allogeneic transplants decreased from 30.6% to 16.6% ($p < 0.0001$). Placental cord blood represented a small proportion of allogeneic transplants over these two time periods (7.0% vs. 6.5% in the early era vs. % in later era, $p = 0.282$).

Indications for first transplants performed are presented in Tables 1 and 2. For allogeneic HCT, acute myeloid leukemia (AML) was the most common indication across both eras (36.7%), followed by ALL (14.2%) and myelodysplastic or myeloproliferative disorders (MDS/MPN) (12%). Allogeneic HCT for AML increased between the early and later eras (32.1% vs. 40.5%, $p < 0.0001$), while HCTs for chronic myeloid leukemia (CML) dropped by approximately half over this time (7.3% early vs. 3.4%, $p < 0.0001$). The most common non-malignant disease for which HCT was undertaken was severe aplastic anemia (4.5%). The proportion of allogeneic HCT for aplastic anemia decreased from 5.4% in the early era to 3.8% in the later era ($p < 0.0001$). In contrast, the use of allogeneic HCT for other non-malignant diseases, including inherited bone marrow diseases and immune system disorders, increased from 5.5% to 8.0% ($p < 0.0001$).

Plasma cell disorders (including multiple myeloma) were the most common indication for autologous transplants over both eras (49.4%), followed by non-Hodgkin (30.8%) and Hodgkin lymphoma (9.7%). The relative contributions for most disease indications for autologous HCT remained similar between both eras (Table 2). Breast cancer was a rare indication for autologous HCT in the earlier era (N = 31), which was no longer observed in the later era (N = 0).

Outcomes were assessed through 5-year OS and 100-day NRM. For adults, 5-year OS after allogeneic HCT was similar between the early and later eras (49% [95% C.I. 47–52%] vs. 49% [95% C.I. 47–51%], $p = 0.83$). Autologous HCT recipients experienced improved 5-year OS in the later era (55% [95% C.I. 54–57%] vs. 65% [95% C.I. 62–67%], $p < 0.0001$) (Figure 1A, Table 3). For pediatric allogeneic HCT, there was significant improvement in 5-year OS in the later era (64% [95% C.I. 60–67%] vs. 76% [95% C.I. 71–80%], $p < 0.0001$), while 5-year OS after pediatric autologous HCT showed a numerical increment, albeit with statistical non-significance (51% [95% C.I. 45–57%] vs. 57% [95% C.I. 49–65%], $p = 0.45$) (Figure 1B, Table 3).

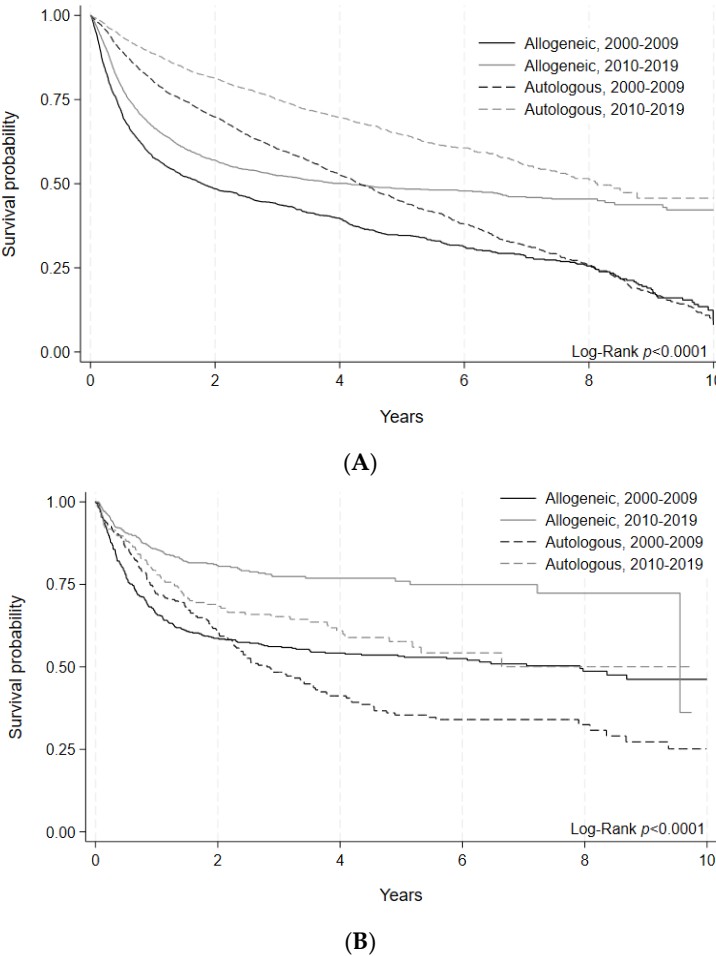

**Figure 1.** (**A**) Overall survival of allogeneic and autologous HCT by era in adult patients. Survival probability indicated by year and transplant type for HCTs performed during each time period. Log-rank ($p < 0.0001$). (**B**) Overall survival of allogeneic and autologous HCT by era in pediatric patients. Survival probability indicated by year and transplant type for HCTs performed during each time period. Log-rank ($p < 0.0001$).

100-day NRM for allogeneic HCT improved for adults in the later era (8.1% [95% C.I. 7.1–9.0%] vs. 10.9% [95% C.I. 9.7–12.2%], $p = 0.0002$). An improvement in 100-day NRM was also noted in pediatric allogeneic HCT patients (3.7% [95% C.I. 2.5–5.3%] vs. 7.5% [95% C.I 5.7–9.6%], $p = 0.0025$) (Figure 2A, Table 4). For autologous HCT, 100-day NRM improved in the later era for adult patients (1.4% [95% C.I. 1.0–1.7%] vs. 2.1% [95% C.I. 1.7–2.6%], $p = 0.0067$), but for pediatric patients, there was no difference in 100-day NRM between the two eras (1.4% [95% C.I. 0.5–3.4%] vs. 1.9% [95% C.I. 0.8–3.8%], $p = 0.534$) (Figure 2B, Table 4). As can be seen in these comparisons, 100-day NRM was consistently higher amongst adults compared to pediatric patients for all transplant modalities.

**Table 3.** Overall survival of allogeneic and autologous HCT by era and transplant type. Adult and pediatric patients receiving a first HCT between 1 January 2000 and 31 December 2019 were categorized by transplant type and time period. Number of individuals at risk and overall survival with 95% confidence intervals are indicated at 1, 2, 5, and 10 years post-transplant.

| | | Allogeneic HCT | | | | | Autologous HCT | | | | | |
| | | 2000–2009 | | 2010–2019 | | | 2000–2009 | | 2010–2019 | | | Overall |
| | | No. at Risk | OS, % (95% CI) | No. at Risk | OS, % (95% CI) | *p*-Value | No. at Risk | OS, % (95% CI) | No. at Risk | OS, % (95% CI) | *p*-Value | *p*-Value |
|---|---|---|---|---|---|---|---|---|---|---|---|---|
| **Adult** | **1 yr** | 1428 | 66 (64–68) | 1621 | 67 (65–69) | 0.1293 | 2838 | 83 (82–85) | 2901 | 88 (87–89) | <0.0001 | |
| | **2 yr** | 1207 | 59 (57–61) | 1050 | 57 (55–59) | 0.9466 | 2278 | 75 (73–76) | 2051 | 82 (80–83) | <0.0001 | |
| | **5 yr** | 757 | 49 (47–52) | 454 | 49 (47–51) | 0.8347 | 1196 | 55 (54–57) | 735 | 65 (62–67) | <0.0001 | <0.0001 |
| | **10 yr** | 458 | 43 (41–45) | 35 | 45 (41–47) | 0.6482 | 556 | 40 (38–42) | 28 | 49 (45–53) | <0.0001 | |
| **Pediatric** | **1 yr** | 459 | 73 (69–76) | 424 | 85 (82–88) | <0.0001 | 234 | 78 (73–82) | 159 | 78 (72–83) | 0.972 | |
| | **2 yr** | 392 | 67 (64–71) | 289 | 80 (77–83) | <0.0001 | 194 | 69 (63–73) | 119 | 68 (61–74) | 0.8041 | |
| | **5 yr** | 299 | 64 (60–67) | 78 | 76 (71–80) | <0.0001 | 121 | 51 (45–57) | 45 | 57 (49–65) | 0.4523 | <0.0001 |
| | **10 yr** | 142 | 61 (57–65) | 2 | 55 (21–78) | <0.0001 | 67 | 47 (41–53) | 1 | 50 (39–60) | 0.612 | |

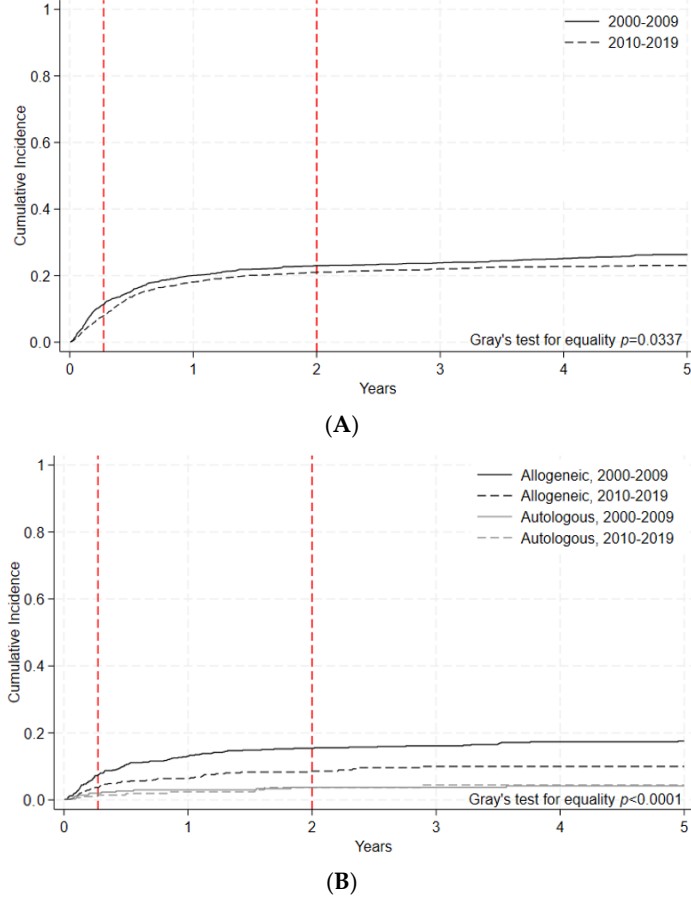

**Figure 2.** (**A**) Nonrelapse mortality (NRM) of adult allogeneic and autologous HCT by era. Cumulative incidence of non-relapse mortality events indicated by year and transplant type during each time period.

Gray's test for equality (*p* < 0.0001). (**B**) Non-relapse mortality of pediatric allogeneic and autologous HCT by era. Cumulative incidence of NRM events indicated by year and transplant type during each time period. Gray's test for equality (*p* < 0.0001).

**Table 4.** Non-relapse mortality (NRM) of allogeneic and autologous HCT by era and transplant type. Adult and pediatric patients receiving a first HCT between 1 January 2000 and 31 December 2019 were categorized by transplant type and time period. Number of individuals at risk and cumulative incidence of NRM with 95% confidence intervals are indicated at 100 days, 1 year, 2 years, and 5 years post-transplant.

| | | Allogeneic HCT | | | | | Autologous HCT | | | | | Overall *p*-Value |
|---|---|---|---|---|---|---|---|---|---|---|---|---|
| | | 2000–2009 | | 2010–2019 | | | 2000–2009 | | 2010–2019 | | | |
| | | No. at Risk | Cumulative Incidence, % (95% CI) | No. at Risk | Cumulative Incidence, % (95% CI) | *p*-Value | No. at Risk | Cumulative Incidence, % (95% CI) | No. at Risk | Cumulative Incidence, % (95% CI) | *p*-Value | |
| Adult | 100 days | 1980 | 10.9 (9.7–12.2) | 2454 | 8.1 (7.1–9.0) | 0.0002 | 3624 | 2.1 (1.7–2.6) | 3479 | 1.4 (1.0–1.7) | 0.0067 | <0.0001 |
| | 1 yr | 1316 | 19.7 (18.1–21.3) | 1501 | 17.9 (16.5–19.4) | 0.0598 | 2443 | 3.9 (3.3–4.5) | 2430 | 2.5 (1.9–2.9) | 0.0002 | |
| | 2 yr | 1099 | 22.3 (20.6–24.0) | 975 | 21.1 (19.5–22.7) | 0.1359 | 1746 | 5.1 (4.3–5.8) | 1598 | 3.1 (2.5–3.7) | <0.0001 | |
| | 5 yr | 690 | 26.5 (24.6–28.4) | 422 | 23.2 (21.5–24.9) | 0.0281 | 822 | 8.3 (7.2–9.3) | 494 | 5.2 (4.2–6.3) | <0.0001 | |
| Pediatric | 100 days | 612 | 7.5 (5.7–9.6) | 587 | 3.7 (2.5–5.3) | 0.0025 | 299 | 1.9 (0.8–3.8) | 205 | 1.4 (0.5–3.4) | 0.5304 | <0.0001 |
| | 1 yr | 443 | 13.0 (10.6–15.6) | 417 | 6.4 (4.6–8.4) | <0.0001 | 203 | 2.9 (1.5–5.2) | 132 | 2.4 (0.9–4.9) | 0.567 | |
| | 2 yr | 384 | 15.4 (12.7–18.2) | 283 | 8.6 (6.4–11.1) | 0.0001 | 158 | 3.7 (2.0–6.8) | 97 | 3.7 (1.6–6.9) | 0.7837 | |
| | 5 yr | 291 | 17.6 (14.7–20.6) | 79 | 9.9 (7.5–12.8) | 0.0001 | 108 | 4.2 (2.3–6.8) | 40 | 4.5 (2.1–8.1) | 0.9081 | |

To better evaluate the changes in clinically relevant outcomes, OS and NRM were separately assessed for non-malignant and select malignant disease indications. For non-malignant disease indications, 5-year OS was numerically improved in the later era for adult patients (85% [95% C.I. 75–90%] vs. 70% [95% C.I. 59–79%], *p* = 0.1529) with a statistically significant improvement for pediatric patients (85% [5% C.I. 78–89%] vs. 78% [95% C.I. 70–84%], *p* = 0.0393) (Figure 3A, Table 5A). 100-day NRM for non-malignant diseases decreased numerically in the later era for adult patients (*p* = 0.1686). For pediatric patients, 100-day NRM similarly showed a downward trend (*p* = 0.2425) between the early and later eras (Figure 3B, Table 5B).

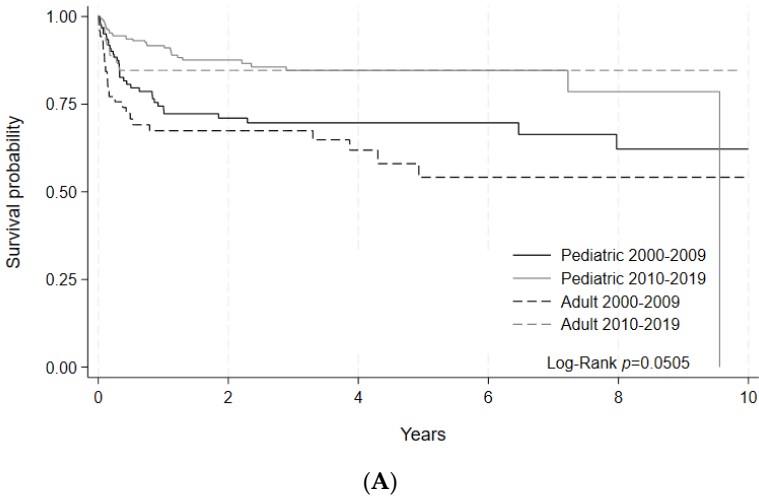

(**A**)

**Figure 3.** *Cont.*

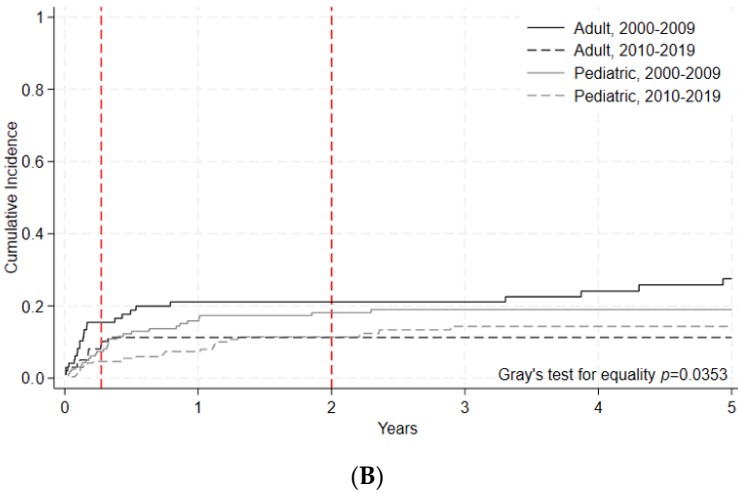

**(B)**

**Figure 3.** (**A**) Overall survival of HCT by era for non-malignant disease indications. Survival probability indicated by year and age group for HCTs performed during each time period. Log-rank (*p* = 0.0505). (**B**) Nonrelapse mortality (NRM)of HCT by era for non-malignant disease indications. Cumulative incidence of NRM events indicated by year and age group for HCTs performed during each time period. Gray's test for equality (*p* = 0.0353).

A subgroup analysis of allogeneic HCTs was performed on adults with indications of acute leukemias, myelodysplastic syndromes, and myeloproliferative neoplasms (AML, ALL, AL of ambiguous lineage, and MDS or MPN), excluding chronic myeloid leukemia. Within this subgroup, 5-year OS was stable: 44.8% [95% C.I 42.1–47.5%] in the early era vs. 45.7 [95% C.I. 43.1–48.2%] in the later era, *p* = 0.1693 (Figure 4A, Table 6A). For these patients, 100-day NRM decreased significantly in the later era (7.85% [95% C.I 6.8–8.9] vs. 11.4% [95% C.I 9.9–13.1], *p* = 0.0002) (Figure 4B, Table 6B).

**Table 5.** (**A**) Overall survival of HCT by era for non-malignant disease indications. Adult and pediatric patients receiving a first HCT for non-malignant disease indications between 1 January 2000 and 31 December 2019 were categorized by transplant type and time period. Number of individuals at risk and overall survival with 95% confidence intervals are indicated at 1, 2, 5, and 10 years post-transplant. (**B**) Nonrelapse mortality (NRM) of HCT by era for non-malignant disease indications. Adult and pediatric patients receiving a first HCT for a non-malignant disease indication between 1 January 2000 and 31 December 2019 were categorized by transplant type and time period. Number of individuals at risk and cumulative incidence of NRM with 95% confidence intervals are indicated at 100 days, 1 year, 2 years, and 5 years post-transplant.

| | | **(A)** | | | | | |
|---|---|---|---|---|---|---|---|
| | | **2000–2009** | | **2010–2019** | | *p*-**Value** | **Overall** *p*-**Value** |
| | | **No. at Risk** | **OS, % (95% CI)** | **No. at Risk** | **OS, % (95% CI)** | | |
| **Adult** | **1 yr** | 66 | 76 (66–84) | 59 | 85 (75–90) | 0.3199 | 0.1529 |
| | **2 yr** | 62 | 76 (66–84) | 47 | 85 (75–90) | 0.3199 | |
| | **5 yr** | 41 | 70 (59–79) | 24 | 85 (75–90) | 0.1529 | |
| | **10 yr** | 27 | 70 (59–79) | 0 | - | 0.1529 | |
| **Pediatric** | **1 yr** | 111 | 81 (74–86) | 165 | 91 (87–95) | 0.0037 | 0.0923 |
| | **2 yr** | 97 | 79 (71–85) | 106 | 87 (82–91) | 0.0153 | |
| | **5 yr** | 77 | 78 (70–84) | 30 | 85 (78–89) | 0.0393 | |
| | **10 yr** | 41 | 75 (67–82) | 2 | 53 (10–83) | 0.0923 | |

**Table 5.** *Cont.*

| | | (B) | | | | | |
|---|---|---|---|---|---|---|---|
| | | 2000–2009 | | 2010–2019 | | *p*-Value | Overall *p*-Value |
| | | No. at Risk | Cumulative Incidence, % (95% CI) | No. at Risk | Cumulative Incidence, % (95% CI) | | |
| Adult | 100 days | 78 | 15.5 (9.0–23.3) | 85 | 9.2 (4.5–15.8) | 0.1686 | 0.0245 |
| | 1 yr | 67 | 21.1 (13.5–29.8) | 60 | 11.3 (5.9–18.4) | 0.0704 | |
| | 2 yr | 63 | 21.1 (13.5–29.8) | 48 | 11.3 (5.9–18.4) | 0.0704 | |
| | 5 yr | 42 | 27.5 (18.3–37.5) | 26 | 11.3 (5.9–18.4) | 0.0245 | |
| Pediatric | 100 days | 142 | 7.4 (4.0–12.1) | 219 | 4.6 (2.4–7.8) | 0.2425 | 0.1434 |
| | 1 yr | 112 | 15.9 (10.5–22.1) | 170 | 7.4 (4.5–11.2) | 0.0109 | |
| | 2 yr | 98 | 18.2 (12.3–24.8) | 107 | 11.4 (7.3–16.4) | 0.0406 | |
| | 5 yr | 78 | 19.0 (13.0–25.8) | 31 | 14.3 (9.4–20.2) | 0.0946 | |

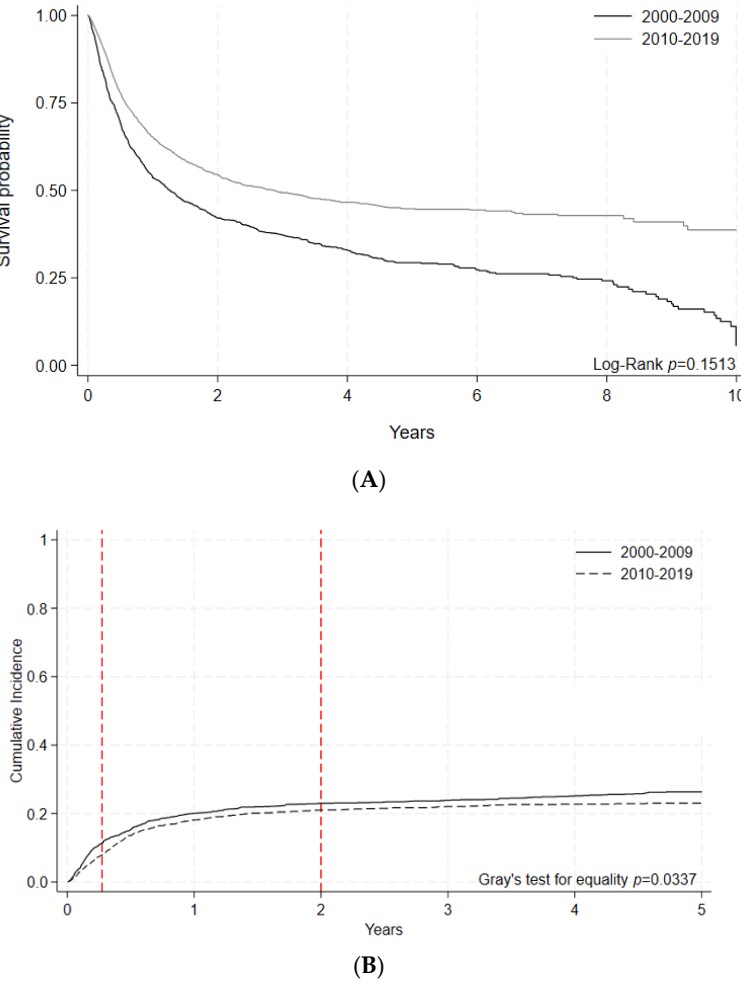

**Figure 4.** (**A**) Overall survival by era following allogeneic HCT indicated for adult acute leukemias. Malignant disease indications included were AML, ALL, AL of ambiguous lineage, and MDS/MPN. Survival probabilities indicated by year for HCTs performed during each time period. Log-rank (*p* = 0.1513). (**B**) Nonrelapse mortality (NRM) by era following allogeneic HCT indicated for adult acute leukemias. Malignant disease indications included were AML, ALL, AL of ambiguous lineage, and MDS/MPN. Cumulative incidence of NRM events indicated by year for HCTs performed during each time period. Gray's test for equality (*p* = 0.0337).

**Table 6.** (**A**) Overall survival by era following allogeneic HCT indicated for adult acute leukemias. Malignant disease indications included were AML, ALL, AL of ambiguous lineage, and MDS/MPN. Number of individuals at risk and overall survival with 95% confidence intervals are indicated at 1, 2, 5, and 10 years post-transplant. (**B**) Nonrelapse mortality (NRM) by era following allogeneic HCT indicated for adult acute leukemias. Malignant disease indications included were AML, ALL, AL of ambiguous lineage, and MDS/MPN. Number of individuals at risk and cumulative incidence of NRM with 95% confidence intervals are indicated at 100 days, 1 year, 2 years, and 5 years post-transplant.

| (A) | | | | | | |
|---|---|---|---|---|---|---|
| | **2000–2009** | | **2010–2019** | | *p*-Value | **Overall *p*-Value** |
| | **No. at Risk** | **OS, % (95% CI)** | **No. at Risk** | **OS, % (95% CI)** | | |
| **1 yr** | 831 | 62.4 (59.8–64.9) | 1222 | 65.6 (63.6–67.6) | 0.014 | 0.1513 |
| **2 yr** | 672 | 53.6 (50.9–56.2) | 762 | 55.0 (52.8–57.3) | 0.0911 | |
| **5 yr** | 423 | 44.8 (42.1–47.5) | 307 | 45.7 (43.1–48.2) | 0.1693 | |
| **10 yr** | 268 | 40.1 (37.3–42.9) | 24 | 41.3 (37.6–44.9) | 0.1522 | |
| (B) | | | | | | |
| | **2000–2009** | | **2010–2019** | | *p*-Value | **Overall *p*-Value** |
| | **No. at Risk** | **Cumulative Incidence, % (95% CI)** | **No. at Risk** | **Cumulative Incidence, % (95% CI)** | | |
| **100 days** | 1216 | 11.4 (9.9–13.1) | 1925 | 7.85 (6.8–8.9) | 0.0002 | 0.0337 |
| **1 yr** | 781 | 20.1 (18.1–22.2) | 1157 | 18.1 (16.5–19.7) | 0.0642 | |
| **2 yr** | 630 | 23.0 (20.8–25.3) | 724 | 21.1 (19.3–22.9) | 0.0815 | |
| **5 yr** | 413 | 26.3 (23.9–28.7) | 293 | 23.0 (21.1–25.0) | 0.0349 | |

## 4. Interpretation

This national-level study provides insights into the trends and outcomes of HCT in Canada over a 20-year period. We observed an increasing number of transplants being offered to Canadians, which is likely to be reflective of improvements in health care access, a greater awareness of the potential benefits of HCT, and a higher number of medically eligible patients as HCT can be delivered more safely. We noted a shift towards older adults receiving both allogeneic and autologous HCTs, with a 5-fold increase in older adults (>64 years old) accessing allogeneic transplants. We also observed that MUDs displaced MRDs as the dominant donor type in allogeneic HCT, suggesting an increased reliance on national and international stem cell donor registries to match Canadian patients.

Regarding key clinical outcomes, there were significant temporal improvements in OS that benefited both pediatric and adult allogeneic HCT recipients. We highlight the ongoing challenge of higher NRM in allogeneic HCT compared to autologous HCT, likely due to deeper and more prolonged immune deficiency after allogeneic HCT, often compounded by GVHD. Age-related differences in NRM after allogeneic HCT were apparent, with higher NRM in adults compared to children, suggesting that adult patients are more vulnerable to the adverse effects of allogeneic HCT. However, there were temporal improvements in NRM in the 2010–2019 (later) era for both adult autologous and allogeneic HCT recipients as well as for children receiving allogeneic HCT.

The rise in both allogeneic and autologous HCT among older adults in the later era is consistent with trends in other jurisdictions [10] and poses unique challenges. While older adults may benefit from allogeneic HCT with appropriate selection [14], they experience inferior outcomes in OS and NRM compared to younger adults [15]. This is due to the associated increased incidence of frailty, medical comorbidities, and inherently more treatment-resistant hematological diseases in older adults [16,17]. We observed a 5-fold increase in allogeneic HCT among older adults during the later era, which potentially impacted OS in these patients. However, there was a measurable reduction in 100-day NRM

in allogeneic HCT patients treated for acute leukemia and related myeloid malignancies, which attests to temporal improvements in both patient selection and the hospital-based care of transplant recipients, despite their older age. Advances in pre- and post-transplant care, such as reduced-intensity conditioning regimens, more effective graft-vs.-host disease prophylaxis, and the optimal use of antimicrobial agents, also likely contributed to these positive changes [18].

In the later era, we observed that MUDs became the dominant donor type for allogeneic HCT, with a proportionate reduction in the use of matched related donors. We also noted a rise in haploidentical (HLA mismatched) related donors. While similar trends have been reported in other registries, the proportion of haploidentical related transplants in Canada remains relatively low compared to other registries [10,11]. Over the study period, cord blood transplant usage remained stable. The shift towards matched unrelated donors may be attributed to decreasing family sizes in Canada [19,20], stem cell registry expansion, and improved GVHD prophylaxis with antithymocyte globulin [21]. We also noted reduced use of MMUDs in the later era, likely reflective of the recognition that these donors were associated with poorer transplant outcomes compared to MUD transplants [22]. However, there is renewed interest in MMUD transplants based on the adoption of post-transplant cyclophosphamide (PTCY) GVHD prophylaxis; this innovation may substantially change trends and outcomes for future transplant recipients [23].

Additionally, the preference for PBSCs over bone marrow as the cell source in the later era highlights improvements in procedural logistics, the relative ease of PBSC collection, and improved donor experiences associated with this apheresis-based collection technique [24].

Our study highlights the persistent challenge of higher NRM in allogeneic HCT compared to autologous HCT. This difference is likely attributable to greater immune suppression associated with allogeneic HCT, as well as the inherent risk of GVHD and its potentially severe complications [25]. We also observed a higher NRM in adult allogeneic recipients compared to pediatric recipients. NRM rises for patients aged 21–40 compared to those under 20 [26]. Similar findings from a national study in South Korea reported increased early NRM following allogeneic HCT in AML patients over 20 years of age compared to younger patients [27]. Moreover, a recent review highlighted studies reporting a lower incidence of severe acute GVHD among pediatric recipients compared to adults [28], with GVHD incidence increasing with age within pediatric cohorts [29]. Taken together, these insights demonstrate age-related differences in HCT outcomes and emphasize the importance of tailored approaches for distinct patient groups.

Amongst all the age groups and time periods that we studied, the outcomes after allogeneic HCT for non-malignant diseases were better than those for patients with malignancies. Amongst non-malignant hematological diseases, the principal indication in Canada was severe aplastic anemia, a disease that is expected to be cured after allogeneic HCT in most patients. As the ancestry of the Canadian population becomes enriched for individuals of Asian and African ancestry, we expect that sickle cell disease and thalassemia may be become more frequent reasons for allogeneic HCT, especially in children and young adults. Future outcome studies should assess trends and outcomes for these emerging conditions.

This study has several limitations. Firstly, this analysis is based on transplants voluntarily reported to the CTTC registry, and thus, it does not account for all HCTs performed in Canada during that time. While CTTC registry data under-represent the entirety of transplant activity in Canada, the potential for geographical bias is low given that individual HCT centres do not specialize in specific patient populations or treatment modalities. Thus, potential under-reporting of HCT data from some centres does not represent a source of differential bias in transplant type or outcome. The quality of national-level registry data depends on the uniform participation of centres that undertake transplants, a systematic and consistent approach to data capture, and a rigorous audit of collected data. Guidelines on the collection and use of "real world data" have been recently disseminated [30]. Although the current registry meets some of these requirements, registry quality may be

further improved by following such guidelines. In addition, the mandatory reporting of essential transplant outcomes by government agencies, such as in the case of the C.W. Bill Young Cell Transplantation Program (CWBYCTP) in the United States, may improve the transparency and quality of clinical care [31].

Regarding outcomes reported in this study, it is important to consider the decreasing reliability of the data with time due to follow-up loss, especially beyond 10 years post-transplant, and the increased likelihood of other unrelated factors influencing OS and NRM. However, transplant centres are generally highly committed to the long-term care of their patients, especially in the allogeneic HCT setting, where follow-up with recipients is often indefinite in duration. This close follow-up of patients ensures a higher quality of long-term outcome data. Additionally, we were unable to report on trends in the use of reduced-intensity or nonmyeloablative conditioning regimens for allogeneic HCT, which gained popularity in the recent era. Despite this limitation, we acknowledge that the increased use of these novel conditioning regimens likely contributed to the observed improvements in outcomes in allogeneic HCT during the later era.

## 5. Conclusions

In conclusion, over the 20-year study period, transplant activity has increased, and key clinical outcomes have generally improved for adult and pediatric populations receiving HCT in Canada. The landscape of HCT in Canada has also evolved to serve older patients, with increased reliance on volunteer unrelated donors. Despite offering allogeneic HCT to older patients, OS rates in adults remained stable. These data serve as a benchmark for quality management in HCT centres across the country and should be helpful for resource planning by health providers and funding authorities.

**Author Contributions:** M.D.S., I.P. and K.P. conceived the project, analyzed the data, searched the literature, and wrote and edited the article. D.S.A., O.B., G.M., O.I. and E.R. created and analyzed the data, performed the statistical analyses, and edited the article. G.P. performed the statistical analyses, searched the literature, and wrote and edited the article. S.B., T.H.T., K.A.H. and A.P. interpreted the data, searched the literature, and edited the article. All authors have read and agreed to the published version of the manuscript.

**Funding:** The CTTC registry is supported by unrestricted research grants from: Kite, a Gilead Company, Bristol Myers Squibb, Medexus, and Sanofi. The CTTC registry is grateful to the Centre for International Bone Marrow Transplant Research (CIBMTR), Milwaukee, WI, for their provision of Canadian transplant centre data to the CTTC registry.

**Institutional Review Board Statement:** This study was approved by the institutional review board of the University of Manitoba (H2022: 253).

**Informed Consent Statement:** Individual patient consent was waived due to the use of anonymous data as reported to and analyzed by the CTTC registry. This approach was deemed acceptable by the University of Manitoba Research Ethics Board.

**Data Availability Statement:** The data that support the findings of this study are not publicly available to ensure and maintain the privacy and confidentiality of individuals' health information. Requests for data may be made to the appropriate data stewards (CancerCare Manitoba's Research and Resource Impact Committee) who may be contacted via the corresponding author.

**Conflicts of Interest:** Hay: ad hoc participation on advisory boards for BMS, Kite, Novartis, and Janssen, as well as research funding from Janssen. Seftel: ad hoc participation on an advisory board for Beigene.

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
