# Peer review of "Hematopoietic Cell Transplantation Trends and Outcomes in Canada: A Registry-Based Cohort Study"

_curroncol, doi:10.3390/curroncol30110723_

Round 1
Reviewer 1 Report
Comments and Suggestions for Authors
Thank you to the authors; no substantial changes requested from my side.
Minors issues identified:
1) Please specify what "mismatched" URD means - 7/8, or less than 7/8? I was surprised to see 5.6% of those for the early period, maybe expand a bit on this trend? Megadose CD34+? I assume those did not use the current PT Cy protocolls
2) "NRM" for non-malignant conditions. Does that mean mortility not related to the underlying condition?
3) 3,4 / 4.9% of products from unknown cell source. Are those randomly distributed on auto/allo and donor type, or is there a potential bias?
Best regards
Author Response
We thank this reviewer for their constructive feedback. Please see responses to each of the reviewer’s suggestions:
1) Please specify what "mismatched" URD means - 7/8, or less than 7/8? I was surprised to see 5.6% of those for the early period, maybe expand a bit on this trend? Megadose CD34+? I assume those did not use the current PT Cy protocolls
Author response: We agree that the definition and characteristics of mismatched URDs (MMUDs) should be articulated further. During the study period (2000-2019) when fully matched ( 8/8 or 10/10) unrelated donors were not identified, it was standard practice at the Canadian centres to offer a single antigen mismatched unrelated donor (7/8 or 9/10), while still using the same conditioning regimen, cell source, and GVHD prophylaxis regimen. Megadose CD34 transplants, ex-vivo T-cell depletion methods, or post-transplant cyclophosphamide were not employed during the study period. In essence, the MMUDs to which the reviewer refers consists of single antigen MMUDs. We have added an explanatory section about this in both the Results and Discussion sections, as follows:
Results section: “Mismatched unrelated donors (MMUD) consisted of single antigen HLA mismatches (7/8 or 9/10, depending on the HLA testing methodology used at each centre). Use of MMUDs dropped in the more recent era (5.6% vs. 0.5%, p<0.0001).”
Discussion Section: “We also noted reduced use of MMUDs in the later era, likely reflective of the recognition that these donors were associated with poorer transplant outcomes compared to MUD transplants[23]. However, there is renewed interest in MMUD transplants based on the adoption of post-transplant cyclophosphamide (PTCY) GVHD prophylaxis; this innovation may substantially change trends and outcomes for future transplant recipients. [24].
2) "NRM" for non-malignant conditions. Does that mean mortality not related to the underlying condition?
Author response: NRM is now more clearly defined in the methods section as follows, “NRM was defined as death from any cause other than underlying disease relapse or progression after HCT”. This definition applies regardless of whether the indication for HCT was for malignant of non-malignant diagnoses.
3) 3,4 / 4.9% of products from unknown cell source. Are those randomly distributed on auto/allo and donor type, or is there a potential bias?
Author response: Based on the reviewer’s astute observation, we re-analyzed the data pertaining to “unknown cell sources for both allogeneic and autologous HCT. The numbers reported in our initial draft were incorrect; we apologize for this. thankfully those transplants with missing ( and thus unknown) cell source was substantially smaller, and did not appear to be biased towards allo vs. auto donor type. Please refer to corrected Tables 1 and 2 for the corrected data. Cell source data was missing in only 42 (0.5%) of allogeneic HCTs. Cell source data was missing in 70 (0.7%) of autologous HCTs.
Reviewer 2 Report
Comments and Suggestions for Authors
The authors comprehensively summarize hematopoietic stem cell transplantation (HSCT) outcomes in Canada over the past 20 years. This is official Canadian data and is a very valuable report.
These data are essential for the promotion of future research on HSCT, and I believe that this report should be accepted for publication in Current Oncology.
Minor comments
It would be more valuable if recommendations and measures for future directions were added to the report. For example, how to collect registry data more reliably for Canada could be added.
Author Response
We thank this reviewer for their constructive feedback. Please see response to the reviewer’s suggestions:
Reviewer comment: It would be more valuable if recommendations and measures for future directions were added to the report. For example, how to collect registry data more reliably for Canada could be added.
Author response: We agree the paper would benefit from further discussion about how registry data could be more reliably collected. Please see the addition in the discussion section, as follows:
“The quality of national level registry data depends on uniform participation of centres that undertake transplants, a systematic and consistent approach to data capture, and rigorous audit of collected data. Guidelines on the collection and use of “real world data” have been recently disseminated [30]. Although the current registry meets some of these requirements, registry quality may be further improved by following such guidelines. In addition, mandatory reporting of essential transplant outcomes by government agencies, such as in the case of the C.W. Bill Young Cell Transplantation Program (CWBYCTP) in the United States may improve the transparency and quality of clinical care [31].
- von Elm E, Altman D.G., Egger M, Pocock S.J., Gotzsche P.C., Vandenbroucke J.P. The Strengthening the Reporting of Observational Studies in Epidemiology (STROBE) Statement: guidelines for reporting observational studies. Lancet. 2007;370(9596):1453-1457.
- C.W Bill Young Transplantation Program. Available Online: https://bloodstemcell.hrsa.gov/about/legislation. (accessed 2023-11-07)”
Round 2
Reviewer 1 Report
Comments and Suggestions for Authors
All my comments have been thoroughly answered.
Thank you
Reviewer 2 Report
Comments and Suggestions for Authors
Appropriate comments are provided in response to the points raised by the reviewer.
These data are important for understanding hematopoietic stem cell transplantation (HSCT) outcomes in Canada, and I look forward to seeing them in print.